# Learning receptive field properties of complex cells in V1

Yanbo Lian[1]*, Ali Almasi[2], David B. Grayden[1], Tatiana Kameneva[1,3], Anthony N. Burkitt[1‡], Hamish Meffin[1,2,4‡]

1 Department of Biomedical Engineering, The University of Melbourne, Melbourne, Australia, 2 National Vision Research Institute, The Australian College of Optometry, Melbourne, Australia, 3 Faculty of Science, Engineering and Technology, Swinburne University, Melbourne, Australia, 4 Department of Optometry and Vision Sciences, The University of Melbourne, Melbourne, Australia

‡ These authors share joint senior authorship.
* yanbo.lian@unimelb.edu.au

**Data Availability Statement:** The code is available at https://github.com/lianyunke/Learning-Receptive-Field-Properties-of-Complex-Cells-in-V1.

**Funding:** This work received funding from the Australian Government, via the Australian Research

## Abstract

There are two distinct classes of cells in the primary visual cortex (V1): simple cells and complex cells. One defining feature of complex cells is their spatial phase invariance; they respond strongly to oriented grating stimuli with a preferred orientation but with a wide range of spatial phases. A classical model of complete spatial phase invariance in complex cells is the energy model, in which the responses are the sum of the squared outputs of two linear spatially phase-shifted filters. However, recent experimental studies have shown that complex cells have a diverse range of spatial phase invariance and only a subset can be characterized by the energy model. While several models have been proposed to explain how complex cells could learn to be selective to orientation but invariant to spatial phase, most existing models overlook many biologically important details. We propose a biologically plausible model for complex cells that learns to pool inputs from simple cells based on the presentation of natural scene stimuli. The model is a three-layer network with rate-based neurons that describes the activities of LGN cells (layer 1), V1 simple cells (layer 2), and V1 complex cells (layer 3). The first two layers implement a recently proposed simple cell model that is biologically plausible and accounts for many experimental phenomena. The neural dynamics of the complex cells is modeled as the integration of simple cells inputs along with response normalization. Connections between LGN and simple cells are learned using Hebbian and anti-Hebbian plasticity. Connections between simple and complex cells are learned using a modified version of the Bienenstock, Cooper, and Munro (BCM) rule. Our results demonstrate that the learning rule can describe a diversity of complex cells, similar to those observed experimentally.

## Author summary

Many cortical functions originate from the learning ability of the brain. How the properties of cortical cells are learned is vital for understanding how the brain works. There are many models that explain how V1 simple cells can be learned. However, how V1 complex

Council Discovery Projects Scheme (Project DP140102947) and the grant AUSMURIB000001 associated with ONR MURI grant N00014-19-1-2571. HM acknowledges funding from the ARC Centre of Excellence for Integrative Brain Function (CE140100007). The funders had no role in study design, data collection and analysis, decision to publish, or preparation of the manuscript.

**Competing interests:** The authors have declared that no competing interests exist.

cells are learned still remains unclear. In this paper, we propose a model of learning in complex cells based on the Bienenstock, Cooper, and Munro (BCM) rule. We demonstrate that properties of receptive fields of complex cells can be learned using this biologically plausible learning rule. Quantitative comparisons between the model and experimental data are performed. Results show that model complex cells can account for the diversity of complex cells found in experimental studies. In summary, this study provides a plausible explanation for how complex cells can be learned using biologically plausible plasticity mechanisms. Our findings help us to better understand biological vision processing and provide us with insights into the general signal processing principles that the visual cortex employs to process visual information.

## Introduction

About 60 years ago, Hubel and Wiesel identified two types of neurons in the primary visual cortex (V1) of cat: simple cells and complex cells [1, 2]. They categorized simple cells as neurons that have receptive fields (RFs) with a spatial structure consisting of distinct light (ON) and dark (OFF) regions. Furthermore, the RFs of simple cells exhibit positive summation within ON and OFF regions and show antagonism between ON and OFF regions. Additionally, it was possible to predict simple cell responses to novel stimuli via linear integration across the ON and OFF regions of the RF. Most simple cells respond strongly to oriented edges or gratings with a preference for a particular orientation (grating stimuli consist of spatially periodic light and dark bands at a given orientation).

In contrast, complex cells exhibit significant nonlinear spatial integration. While they respond strongly to moving oriented edges, they do not show the other characteristics of simple cells described above. One important property of complex cells is their spatial phase invariance; i.e., strong responses are evoked by oriented gratings with the preferred orientation, but for a wide range of spatial phases. This distinguishes them from simple cells, which are selective to spatial phase. Spatial phase refers to the position of the light and dark bands in the grating within one periodic cycle. Spatial phase invariance is similar to shift invariance or position invariance, which means that the response is generally not sensitive to the relative position of the stimulus within the RF of a complex cell.

Simple cell responses can be described phenomenologically using a linear-nonlinear model that has three stages [3–5]. First, the input image is linearly filtered by a single spatial filter whose weights represent the cell's RF. Second, the output of the first stage, the feature contrast, is passed through a static nonlinearity (usually one-sided, such as half-wave rectification) to obtain the spike rate. Third, spike trains are generated via a Poisson process with this instantaneous spike rate. This is referred to as the linear-nonlinear-Poisson model [6].

A somewhat similar model with linear-nonlinear processing can be used to describe responses of complex cells. However, for complex cells, multiple linear spatial filters are typically required, and the output of each filter is passed through a nonlinear function that is usually double-sided (e.g., squaring nonlinearity). Following this, the signals are combined across filter channels by summation and can be passed through a final static nonlinearity to give the predicted spike rate [4, 5, 7]. One classical phenomenological model for complex cells is the energy model [4, 8], in which the complex cell responses are the sum of the squared outputs of two linear spatially-phase-shifted filters, as shown in Fig 1A. For the energy model, the squaring function imparts polarity invariance (non-selective to the polarity of the stimulus), and filters with different spatial phases generate spatial phase invariance. Although the energy model

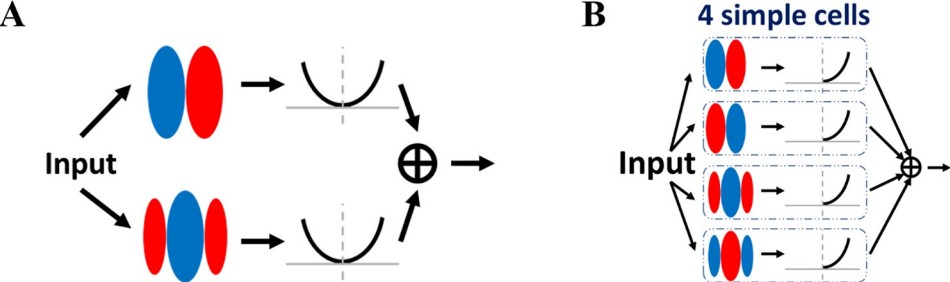

**Fig 1. The energy model.** Red and blue represent ON and OFF sub-regions of the receptive field of a simple cell, respectively. (A) The classic energy model of a complex cell. The linear response of the input convolved with the filter is passed to a two-sided nonlinear function (a power function in this case). The outputs of two nonlinear functions are then summed to generate the response for the complex cell. (B) The equivalent hierarchical structure of the energy model of a complex cell. The response of the complex cell sums over the responses of simple cells.

can capture the spatial phase invariance of some complex cells, recent experimental studies showed that many complex cells in cat visual cortex have spike rates that show large modulations to spatial phase that are not accounted for by the energy model [9–12].

These types of models only describe the responses of neurons to visual stimuli and cannot explain the biophysical mechanisms of complex cell responses. Several models have also investigated different network architectures; these can be divided into three categories: hierarchical, parallel, and recurrent (see [13] for a review). The notion of the hierarchical model was proposed by Hubel and Wiesel [2], where a complex cell pools the activities of simple cells with the same orientation preference but with different spatial phase preferences so that it is orientation selective but spatially phase invariant. This idea was later supported by an experimental study [7]. From the perspective of a hierarchical structure, the energy model can be understood as a complex cell having convergent inputs from four simple cells with different spatial phase preferences. For example, the response of the complex cell is the weighted sum of four simple cells, as shown in Fig 1B.

The concept of the hierarchical structure has been challenged by parallel and recurrent models. In the parallel model [14], it is proposed that both simple and complex cells are generated by separate thalamocortical pathways in parallel. This is supported by the discovery that some complex cells receive direct input from the thalamus [14]. However, some previous studies showed that most complex cells do not receive direct input from the thalamus [15–17]. The idea of the recurrent model is supported by experimental evidence that cortical cells mainly receive most of their input from other cortical neurons instead of the thalamus [18]. Therefore, the response of cortical cells should be primarily determined by recurrent cortical inputs [16]. Nevertheless, thalamocortical connections have many features to make them strong although they only account for a small fraction of excitatory synapses made by cortical cells [13].

In this paper, we investigate how the RFs of complex cells can be learned in such a way that they explain experimental data for complex cells. We focus on the hierarchical structure between simple and complex cells. Namely that complex cells pool feedforward input from simple cells, as supported by experimental results [19]. Here, we address the question of how complex cells learn which simple cells to pool. Given simple cells with RFs of different orientation and spatial phase preferences, there are two distinct possible mechanisms of pooling simple cells to construct complex cells: indiscriminate pooling and selective pooling.

*Indiscriminate pooling* means pooling randomly from any simple cells in a local region of the cortical surface. If simple cells with similar orientation but different spatial phases are

located in a local region via an orientation map, complex cells could be orientation-selective but spatial-phase-invariant by indiscriminate pooling of these simple cells.

*Selective pooling* means pooling simple cells according to certain criteria. For example, only simple cells with RFs of the same orientation but different spatial phase preferences can be pooled.

The orientation maps necessary to maintain orientation selectivity under indiscriminate pooling are prevalent among monkey, cat, and ferret V1 [20–22], and some models were designed to describe complex cells based on the orientation topography of simple cells [23–25]. However, there are also rodent species, such as mouse and rat, that do not have orientation maps but still have complex cells in V1 [26, 27]. Therefore, selective pooling seems to be a more general principle for constructing complex cells by pooling simple cells.

Nevertheless, the question of how synaptic plasticity can selectively pool simple cell inputs with appropriate weights for cells with different orientation and spatial phase selectivities still remains. While some studies have addressed this issue [28–31], most existing models overlook many details of biological reality. Some common problems of current models of complex cells in this regime are listed as follows.

First, many models assume that the nonlinear function applied to filter outputs is two-sided, i.e. the function increases away from zero in both the positive and negative directions of filter output. However, biological simple cells, which form the inputs to the complex cell, have a one-sided spiking nonlinearity. This artificially builds in polarity invariance to the complex cell model, and contributes significantly to spatial phase invariance in an artificial way. This problem exists in the Independent Subspace Analysis (ISA) model designed by Hyvärinen and Hoyer [28] and the Slow Feature Analysis (SFA) model designed by Berkes and Wiskott [29]. These models do not explain how simple cells with similar orientation tuning, but opposite polarity selectivity are pooled via the learning process.

Second, the weights connecting simple and complex cells are not learned in some models. The weights in the ISA model [28] are fixed, with only the weights of the simple cells learned. The weights in Hosoya and Hyvärinen's model [30] are computed by strong dimensionality reduction using Principal Component Analysis (PCA), which does not correspond to a form of synaptic plasticity.

Third, the learning process of some models incorporates artificial components that do not have direct biological realization: the SFA model [29] solves an optimization problem and implies no Hebbian synaptic plasticity and the model designed by Einhäuser et al. [31] only allows one winner neuron to learn in each iteration. Additionally, for the model of Einhäuser et al. [31], the ratio of simple to complex cells, 60: 4, is inconsistent with the experimental evidence that complex cells are at least as prevalent as simple cells in V1 [32]. Therefore, investigating how complex cell properties can be learned through biologically plausible plasticity rules is an open, but important, problem for understanding how the brain works.

One candidate mechanism to solve this problem is efficient coding, which can be implemented in a biologically plausible fashion, through Hebbian plasticity, to explain many experimental phenomena of simple cells [33]. Though efficient coding can learn simple cells, we found that a cascaded stage of efficient coding cannot effectively learn the RF properties of complex cells from simple cell responses (see Discussion and S2 Appendix for details).

In this paper, we propose a biologically plausible model of complex cells based on the Bienenstock, Cooper, and Munro (BCM) synaptic plasticity rule [34, 35] and show that this leads to a model of complex cells that can pool simple cells with various spatial phase preferences. The pooled simple cells form the *subspace* of the complex cell and each pooled simple cell is a *subunit* in the subspace. The learned subspace can account for the spatial phase invariance of

experimentally recorded complex cells. Further analysis of model complex cells demonstrates that the proposed model can account for the diversity of RF properties of complex cells found in a recent experimental study [11].

# Materials and methods

## Structure of the model

The proposed three-layer network of rate-based neurons models the activities of lateral geniculate nucleus (LGN) cells (first layer), V1 simple cells (middle layer), and V1 complex cells (top layer), as shown in Fig 2.

A summary of the parameters of the model that will be used throughout this paper is given in Table 1.

The bottom two layers implement a two-layer model that is biologically plausible and has been previously shown to account for many experimental phenomena [33]. This two-layer model is a variant of sparse coding and incorporates many biological constraints. The dynamics of LGN cells and simple cells are described by the evolution of the membrane potentials, $\mathbf{v}$, followed by application of a threshold-linear function to give firing rates, $\mathbf{r}$:

$$\tau_{\mathrm{L}} \dot{\mathbf{v}}^{\mathrm{L}} = -\mathbf{v}^{\mathrm{L}} + \mathbf{x}^{\mathrm{L}} + (\mathbf{A}^{\mathrm{d},+} + \mathbf{A}^{\mathrm{d},-})\mathbf{r}^{\mathrm{S}} + r_{\mathrm{b,L}},$$

$$\mathbf{r}^{\mathrm{L}} = \max(\mathbf{v}^{\mathrm{L}}, 0),$$

$$(1)$$

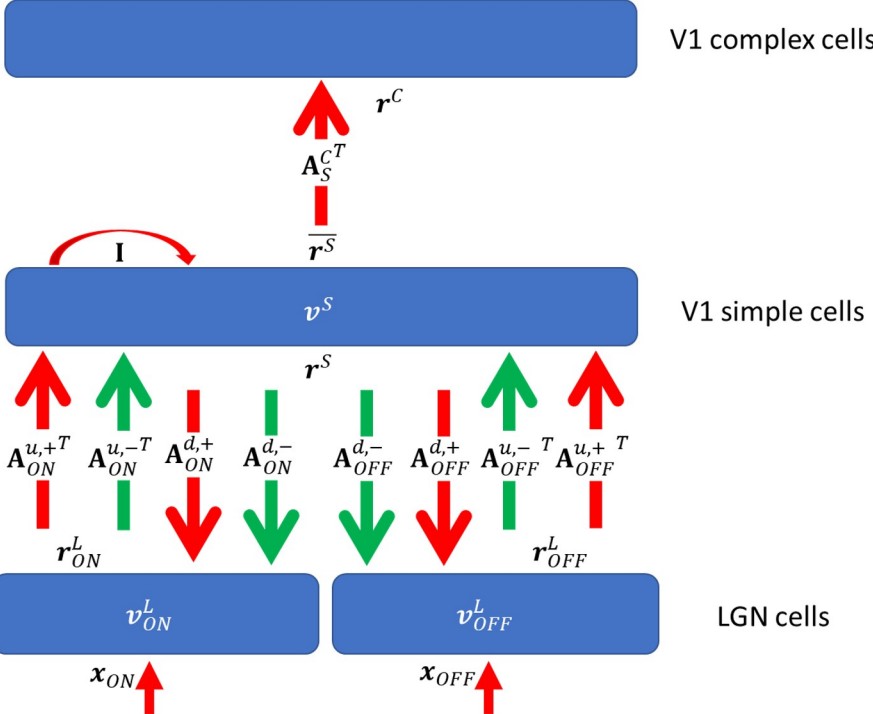

**Fig 2. Graphical representation of the model. I** is the identity matrix that represents self-excitation. Red and green arrows represent excitatory and inhibitory connections, respectively. Upward and downward arrows represent feedforward and feedback pathways. Parameters are defined in Table 1.

**Table 1. Model symbols and parameters.**

| Description | Symbol |
|---|---|
| Input stimuli to LGN/complex cells | $\mathbf{x}^L$ / $\mathbf{x}^C$ |
| Input stimuli to ON/OFF LGN cells | $\mathbf{x}^L_{ON}$ / $\mathbf{x}^L_{OFF}$ |
| Membrane time constants of LGN/simple cells (both 10 ms) | $\tau_L$ / $\tau_S$ |
| Membrane potentials of LGN/simple cells | $\mathbf{v}^L$ / $\mathbf{v}^S$ |
| Membrane potentials of ON/OFF LGN cells | $\mathbf{v}^L_{ON}$ / $\mathbf{v}^L_{OFF}$ |
| Firing rates of LGN/simple/complex cells | $\mathbf{r}^L$ / $\mathbf{r}^S$ / $\mathbf{r}^C$ |
| Firing rates of ON/OFF LGN cells | $\mathbf{r}^L_{ON}$ / $\mathbf{r}^L_{OFF}$ |
| Spontaneous firing rate of LGN (0.5 Hz) | $r_{b,L}$ |
| Leakage voltages of simple cells | $\mathbf{v}^S_{leak}$ |
| Excitatory connection: all LGN cells to simple cells | $\mathbf{A}^{u,+}$ |
| Excitatory connection: ON/OFF LGN cells to simple cells | $\mathbf{A}^{u,+}_{ON}$ / $\mathbf{A}^{u,+}_{OFF}$ |
| Inhibitory connection: all LGN cells to simple cells | $\mathbf{A}^{u,-}$ |
| Inhibitory connection: ON/OFF LGN cells to simple cells | $\mathbf{A}^{u,-}_{ON}$ / $\mathbf{A}^{u,-}_{OFF}$ |
| Excitatory connection: simple cells to all LGN cells | $\mathbf{A}^{d,+}$ |
| Excitatory connection: simple cells to ON/OFF LGN cells | $\mathbf{A}^{d,+}_{ON}$ / $\mathbf{A}^{d,+}_{OFF}$ |
| Inhibitory connection: simple cells to all LGN cells | $\mathbf{A}^{d,-}$ |
| Inhibitory connection: simple cells to ON/OFF LGN cells | $\mathbf{A}^{d,-}_{ON}$ / $\mathbf{A}^{d,-}_{OFF}$ |
| Excitatory connection: simple cells to complex cells | $\mathbf{A}^C_S$ |
| Sparsity level (0.1) | $\lambda_S$ |
| Upper bounds of LGN-simple/simple-complex connection weights (0.3 and 1) | $a_{1,max}$ / $a_{2,max}$ |
| Learning rate of LGN-simple connection weights (3) | $\eta_1$ |
| Learning rates of weights and thresholds for complex cells (both $10^{-3}$) | $\eta_a$ / $\eta_\theta$ |
| Weight regularization constants of LGN-simple/simple-complex connections ($10^{-3}$ and $10^{-4}$) | $\gamma_1$ / $\gamma_a$ |
| Parameters of normalization of complex cells (0.01 and 12) | $\alpha$ / $\beta$ |

and

$$\tau_S \dot{\mathbf{v}}^S = -(\mathbf{v}^S - \mathbf{v}^S_{leak}) + \mathbf{A}^{u,+\,T}_{ON} \mathbf{r}^L_{ON} + \mathbf{A}^{u,-\,T}_{ON} \mathbf{r}^L_{ON}$$
$$+ \mathbf{A}^{u,+\,T}_{OFF} \mathbf{r}^L_{OFF} + \mathbf{A}^{u,-\,T}_{OFF} \mathbf{r}^L_{OFF} + \mathbf{r}^S, \tag{2}$$
$$\mathbf{r}^S = \max(\mathbf{v}^S - \lambda_S, 0),$$

where $\mathbf{x}^L = [\mathbf{x}^{L\,T}_{ON}, \mathbf{x}^{L\,T}_{OFF}]^T$, $\mathbf{v}^L = [\mathbf{v}^{L\,T}_{ON}, \mathbf{v}^{L\,T}_{OFF}]^T$, $\mathbf{r}^L = [\mathbf{r}^{L\,T}_{ON}, \mathbf{r}^{L\,T}_{OFF}]^T$, $\mathbf{A}^{u,+} = [\mathbf{A}^{u,+}_{ON}\ \mathbf{A}^{u,+}_{OFF}]$, $\mathbf{A}^{u,-} = [\mathbf{A}^{u,-}_{ON}\ \mathbf{A}^{u,-}_{OFF}]$, $\mathbf{A}^{d,+} = [\mathbf{A}^{d,+}_{ON}\ \mathbf{A}^{d,+}_{OFF}]$, $\mathbf{A}^{d,-} = [\mathbf{A}^{d,-}_{ON}\ \mathbf{A}^{d,-}_{OFF}]$, $\tau_L$ and $\tau_S$ are the time constants of the membranes of LGN cells and simple cells, $r_{b,L}$ is the background firing rate for LGN cells, and $\lambda_S$ is the threshold of the rectifying function of firing rates. $\mathbf{v}^S_{leak}$ represents the change of membrane potential caused by leakage currents. Details of the bottom two layers can be found in [33].

The response of the complex cells is simply the linear summation of simple cells to which they connect, as given by

$$\mathbf{r}^C = \mathbf{A}^{C\,T}_S \mathbf{x}^C, \tag{3}$$

where $\mathbf{A}^C_S$ is a non-negative matrix that represents excitatory connections between simple and complex cells and $\mathbf{x}^C$ is the input to the complex cell.

The model was simulated in MATLAB (R2020a, USA) and run on the high performance computing platform Spartan [36]. The code is available at https://github.com/lianyunke/Learning-Receptive-Field-Properties-of-Complex-Cells-in-V1.

## Input

**Natural images.** For learning simple cells, the data set used in this paper consists of 50 randomly selected 1024 × 1536 pixel images of calibrated natural scenes from van Hateren's dataset [37].

**Natural video.** For natural visual stimuli with temporal information, such as videos, the content between subsequent frames in a fixed region over a short time period is very similar except for some translations or shifts in position, as results from the movement of an object or self movement. Therefore, the temporal information in a natural video is similar to sequences of translated images, as used to investigate temporal slowness learning [29]. In this paper, we use a natural video to learn the complex cells. However, similar complex cells can also be learned using a sequence of natural images (van Hateren's dataset) with a random spatial jitter to incorporate temporal information (see S1 Appendix for detail).

After simple cells are learned, a 2-minute natural video of size 1080 × 1920 pixels with frame rate 24 Hz is used to learn the connection between simple and complex cells. For this study, the central 800 × 800 pixel region of the video is used (https://youtu.be/K-Vr2bSMU7o, with permission from the owner). In each iteration of the learning process, $N$ consecutive frames of the video are used. For the $N$ chosen frames, the bottom two-layer model generates $N$ sets of simple cell responses. Since the $N$ consecutive frames contain similar features with spatial shifts or rotations, the neural activities of simple cells in response to the $N$ frames will also contain spatial phase information. Using the concept of the trace rule, where the response is determined by the current and past responses [38], the average of $N$ sets of simple cell responses is then used as the input to the complex cells; i.e., $\mathbf{x}^C = \langle \mathbf{r}^S \rangle$. In our opinion, simple cells with similar orientation tuning will have similar average responses to the $N$ video frames, which helps with translation invariance if a suitable learning rule can pool co-active inputs.

**Pre-processing of natural stimuli.** The input patches are first whitened to mimic retinal processing before visual processing. This process involves filtering the input image to model the response of ganglion cells whose RFs are characterized as divisively normalized difference-of-Gaussian filters [39, 40]. The whitened pixel intensity ($I$) at point ($x$, $y$) is calculated by

$$I(x, y) = \frac{I_0(x, y) - I_1(x, y)}{I_d(x, y)}, \tag{4}$$

where $I_0$, $I_1$, and $I_d$ are the outputs from three unit-normalized Gaussian filters: center filter ($g_0$), surround filter ($g_1$), and divisive normalization filter ($g_d$), where $g_d$ captures the local adaptation of ganglion cells [41]. Images are convolved with $g_0$, $g_1$, and $g_d$ to give a 2D retinotopic representation. The standard deviation of the center filters is set to 1 pixel. The standard deviation of surround filters is chosen to be 1.5 pixels or 1.5 times the center filter standard deviation, which is consistent with previous measurements [42]. The standard deviation of $g_d$ has the same size as $g_1$ [40].

The whitened images are then multiplied with a 2D Gaussian windowing filter with standard deviations of 3 pixels in both vertical and horizontal directions. The purpose of this step is to limit the spatial range of synaptic connectivity with LGN and, subsequently, V1 cells, similar to the approach used by Linsker [43]. It also puts more emphasis on the central part of the image patch to cause the learned simple RFs to be more centralized in the 2D image domain.

This step also assumes that model complex cells pool local simple cells that have RFs in the same region.

Pre-processed image patches of size $M \times M$ are then fed into LGN cells in the first layer. If the pixel intensity is positive, it is set as the input to the ON LGN cell and the input to the corresponding OFF LGN cell is set to zero. If the pixel intensity is negative, the absolute value of the intensity is set as the input to the OFF LGN cell and the input to the corresponding ON LGN cell is set to zero.

## Learning LGN-simple cell connections

Connections between LGN and simple cells are learned based on Hebbian or anti-Hebbian plasticity, the changes of synaptic weights depend only on pre- and post-synaptic activities. The learning rule derives from efficient coding and is similar to [33], given by

$$
\begin{aligned}
\Delta \mathbf{A}^{\mathrm{u},+} &= \eta_1(\langle (\mathbf{r}^{\mathrm{L}} - r_{\mathrm{b,L}})\mathbf{r}^{\mathrm{S}^T}\rangle - \gamma_1 \mathbf{A}^{\mathrm{u},+}), \\
\Delta \mathbf{A}^{\mathrm{u},-} &= \eta_1(\langle (\mathbf{r}^{\mathrm{L}} - r_{\mathrm{b,L}})\mathbf{r}^{\mathrm{S}^T}\rangle - \gamma_1 \mathbf{A}^{\mathrm{u},-}), \\
\Delta \mathbf{A}^{\mathrm{d},+} &= -\eta_1(\langle (\mathbf{r}^{\mathrm{L}} - r_{\mathrm{b,L}})\mathbf{r}^{\mathrm{S}^T}\rangle - \gamma_1 \mathbf{A}^{\mathrm{d},+}), \\
\Delta \mathbf{A}^{\mathrm{d},-} &= -\eta_1(\langle (\mathbf{r}^{\mathrm{L}} - r_{\mathrm{b,L}})\mathbf{r}^{\mathrm{S}^T}\rangle - \gamma_1 \mathbf{A}^{\mathrm{d},-}),
\end{aligned}
\tag{5}
$$

where $\eta_1$ is the learning rate, $\langle \cdot \rangle$ is the ensemble-average operation over a batch of image samples, $\mathbf{r}^{\mathrm{L}} - r_{\mathrm{b,L}}$ is the vector of LGN firing rates $\mathbf{r}^{\mathrm{L}}$ reduced by the spontaneous rate $r_{\mathrm{b,L}}$, $(\mathbf{r}^{\mathrm{L}} - r_{\mathrm{b,L}})\mathbf{r}^{\mathrm{S}^T}$ is the Hebbian matrix given by the outer product of firing rate vectors $\mathbf{r}^{\mathrm{L}} - r_{\mathrm{b,L}}$ and $\mathbf{r}^{\mathrm{S}}$ for the LGN and simple cells, respectively, and $\gamma_1$ is the weight regularization constant that prevents weights from growing without bound. Note that changes in weight in the feedforward direction (LGN to simple cell; superscript $u$) are positively signed and are thus Hebbian, while those in the feedback direction (simple cell to LGN; superscript $d$) are signed negative and are thus anti-Hebbian. The excitatory weights, $\mathbf{A}^{\mathrm{u},+}$ and $\mathbf{A}^{\mathrm{d},+}$, are kept non-negative, while inhibitory weights, $\mathbf{A}^{\mathrm{u},-}$ and $\mathbf{A}^{\mathrm{d},-}$, are kept non-positive during learning. In addition, the absolute value of each weight is limited to an upper bound, $a_{1,\mathrm{max}}$, that represents the maximal synaptic efficacy. The only difference between the previous learning rules of simple cells [33] and this study is that weight normalization is replaced by the combination of self-regularization terms in Eq 5 and the upper bound of connection weights.

## Learning rule for simple-complex cell connections

Connections between simple and complex cells are learned based on the Bienenstock, Cooper, and Munro rule, a form of Hebbian plasticity where the sign and efficacy change depend not only on pre- and post-synaptic activities but also on the slowly varying values of the history of post-synaptic activities [34].

**Bienenstock, Cooper, and Munro (BCM) rule.** The Bienenstock, Cooper, and Munro (BCM) rule [34, 35] can learn underlying features from the input through a competitive process between synapses of a neuron that arises from the thresholding mechanism that is part of the BCM learning rule. However, BCM plasticity is designed for single neurons and does not introduce any competition between network neurons that receive the same visual input. Law and Cooper applied the BCM plasticity rule to a network using natural images as input stimuli, and showed that this learning rule can learn RFs like those of simple cells [44]. However, since the BCM plasticity rule is the same for every neuron in the network, the learned features of the network tend to be similar [45]. By incorporating response normalization, where the response

of a cell is normalized by the responses of other cells in the network [46], a "soft" form of competition is introduced to the network. Furthermore, strong experimental evidence has been found that normalization operates throughout the visual system, from the retina to the visual cortex (see [47] for a review). By incorporating BCM plasticity and normalization, Willmore et al. showed that normalized BCM (NBCM) can learn different simple cell-like RFs when the model is trained on natural images [45]. However, the BCM and NBCM plasticity rules ignore some biological constraints such as Dale's Law [48] and non-negative neuronal responses.

For the synaptic weights between simple cell $i$ and complex cell $j$, $a_{i,j}$, the BCM learning rule updates the weight according to not only pre- and post-synaptic activities, $x_i^C$ and $r_j^C$, but also a learned threshold for complex cell $j$, $\theta_j$,

$$
\begin{aligned}
\Delta a_{i,j} &= \eta_a x_i^C r_j^C (r_j^C - \theta_j), \\
\Delta \theta_j &= \eta_\theta ((r_j^C)^2 - \theta_j),
\end{aligned}
\tag{6}
$$

where $\eta_a$ and $\eta_\theta$ are the learning rates that determine the rates of change of the synaptic weight and threshold. The original BCM rule allows weight $a_{i,j}$ to change signs, which is not biologically plausible. Modified rules based on BCM are introduced below to incorporate biological constraints. Note that the original BCM rule is given here for completeness (Eq 6), but only the modified BCM and NBCM rules, given below, were used in this study.

**Modified BCM rule.** For the modified BCM rule, the synaptic weight between simple cell $i$ and complex cell $j$, $a_{i,j}$, is updated by the learning rule,

$$
\begin{aligned}
\Delta a_{i,j} &= \eta_a (x_i^C r_j^C (r_j^C - \theta_j) - \gamma_a a_{i,j}), \\
\Delta \theta_j &= \eta_\theta ((r_j^C)^2 - \theta_j),
\end{aligned}
\tag{7}
$$

where $\gamma_a$ is the weight regularization constant. In addition, the connections between simple and complex cells are excitatory in the model, so $a_{i,j}$ is kept non-negative during learning. The upper bound of the connection weights is explicitly constrained to be $a_{2,\max}$.

Note that the modified BCM learning rule (Eq 7) differs from the original BCM learning rule (Eq 6) in three ways. First, the original BCM rule allows weights to change signs, which is not permitted in the modified BCM rule. Second, $a_{i,j}$ is constrained by the maximal weight, $a_{2,\max}$. Third, the original BCM rule does not have the weight regularization term, $-\gamma_a a_{i,j}$; this term is added to prevent weights from growing without bound and to push unimportant weights to zero.

**Modified NBCM rule.** The original NBCM rule proposed by Willmore et al. [45] shows that this learning rule can learn different RFs for neurons in a network. NBCM incorporates the response normalization model proposed by Heeger [46]. For the model, the response of complex cells, $r_j^C$, is normalized and the normalized response, $r_{j,N}^C$, is then used to update the synaptic weight and threshold, as given by the NBCM learning rule,

$$
\begin{aligned}
r_{j,N}^C &= \frac{\beta r_j^C}{\alpha + \sqrt{\sum_k (r_k^C)^2}}, \\
\Delta a_{i,j} &= \eta_a x_i^C r_{j,N}^C (r_{j,N}^C - \theta_j), \\
\Delta \theta_j &= \eta_\theta ((r_{j,N}^C)^2 - \theta_j),
\end{aligned}
\tag{8}
$$

where the sum over $k$ represents all complex cells in the network and $\alpha$ and $\beta$ are constants that determine the strength of the normalized response, $r^{\mathrm{C}}_{j,N}$, compared with response, $r^{\mathrm{C}}_j$.

The modified NBCM learning rule introduces more constraints on the weights and is given by

$$
\begin{aligned}
r^{\mathrm{C}}_{j,N} &= \frac{\beta r^{\mathrm{C}}_j}{\alpha + \sqrt{\sum_k (r^{\mathrm{C}}_k)^2}}, \\
\Delta a_{i,j} &= \eta_a (x^{\mathrm{C}}_i r^{\mathrm{C}}_{j,N}(r^{\mathrm{C}}_{j,N} - \theta_j) - \gamma_a a_{i,j}), \\
\Delta \theta_j &= \eta_\theta ((r^{\mathrm{C}}_{j,N})^2 - \theta_j),
\end{aligned}
\tag{9}
$$

where $\gamma_a$ is the weight regularization constant. Additionally, the maximal value of the connection weights is explicitly constrained to be $a_{2,\max}$. As above, the modified NBCM learning rule (Eq 9) differs from the original NBCM learning rule (Eq 8) in three ways. First, $a_{i,j}$ is kept non-negative during learning. Second, $a_{i,j}$ is constrained by the maximal connection weight, $a_{2,\max}$. Third, there is a weight regularization term.

It should be mentioned that the normalization equation in Eq 9 uses the global information of other neuron activities to calculate normalized responses of the post-synaptic neuron. This is still consistent with the Hebbian principle of plasticity depending only on pre- and postsynaptic activity as the activity of the postsynaptic neuron is the normalized form of the activity. Such normalization is consistent with experimental data [47]. Mechanistically, this normalization can arise in a biologically plausible fashion through lateral recurrent connections in V1 with physiologically realistic neural dynamics described by a supralinear stabilized network model [49].

## Training

After pre-processing the natural stimuli, input patches of size $16 \times 16$ ($M = 16$) are used in our model, similar to previous studies [33, 50, 51], resulting in 256 ON and 256 OFF LGN cells. We use 100 simple cells and 100 complex cells in the second and third layers, respectively. The membrane time constants, $\tau_{\mathrm{L}}$ and $\tau_{\mathrm{S}}$, for LGN and simple cells are both taken to be 10 ms, which is physiologically plausible [52]. The spontaneous firing rate, $r_{\mathrm{b,L}}$, for LGN cells is chosen to be 0.5 Hz although different values lead to similar results because the spontaneous firing rates only provide a working point for the dynamical model [33]. The dynamical system of the model described by Eqs 1–3 is numerically solved using the first-order Euler method. The system was evolved for 20 iteration steps with integration time step of 4 ms, for calculating the responses for both simple and complex cells. This allowed convergence to a numerically stable solution.

**Simple cells.** The bottom two layers of the network are trained first on the natural image data set (van Hateren's dataset [37]). Since, during the course of training, $\mathbf{A}^{\mathrm{u,+}}$ approaches $-\mathbf{A}^{\mathrm{d,-}}$ and $\mathbf{A}^{\mathrm{u,-}}$ approaches $-\mathbf{A}^{\mathrm{d,+}}$ [33], we simply set $\mathbf{A}^{\mathrm{u,+}} = -\mathbf{A}^{\mathrm{d,-}}$ and $\mathbf{A}^{\mathrm{u,-}} = -\mathbf{A}^{\mathrm{d,+}}$ at the beginning of training. The upper bound of connection weights between LGN and simple cells, $a_{1,\max}$, is set to 0.3 so that the excitatory weights cannot exceed 0.3 and inhibitory weights cannot be less than −0.3. The sparsity level of simple cells, $\lambda_{\mathrm{S}}$, is set to 0.1. In addition, the learning rule (Eq 5) uses a batch that contains 100 randomly selected $16 \times 16$ image patches that have no temporal information in each epoch to accelerate the learning process, similar to previous studies [51, 53]. The weight regularization constant, $\gamma_1$, is a small constant and set to $10^{-3}$ in this study. The learning rate, $\eta_1$, is 3. $10^5$ epochs are used in the training process.

After the training process for simple cells, the connections between LGN and simple cells, $\mathbf{A}^u$ and $\mathbf{A}^d$, are fixed. Then the connections between simple cells and complex cells are trained on the natural video using the learning rules given below.

**Applying the modified BCM rule for complex cells.**   There are $4 \times 10^6$ epochs in the training process because the learning rates for weights and threshold are small, taken to be $\eta_a = 10^{-3}$ and $\eta_\theta = 10^{-3}$, respectively. The weight regularization constant, $\gamma_a$, is set to $10^{-4}$. The maximal connection weight, $a_{2,max}$, is 1, so the connection strength between simple and complex cells indicates how strongly a simple cell is pooled by a complex cell. The number of video frames used in each iteration is $N = 15$. Since the input to the complex cells, $\mathbf{x}^C = \langle \mathbf{r}^S \rangle$, is very small by averaging over the $N$ video frames, $\mathbf{x}^C$ is scaled up by 10 when applying the modified BCM rule.

**Applying the modified NBCM rule for complex cells.**   The Same parameter values as above are used for the modified NBCM rule where possible. There are $4 \times 10^6$ epochs in the training process. The learning rates for weights and threshold are $\eta_a = 10^{-3}$ and $\eta_\theta = 10^{-3}$, respectively. The weight regularization constant, $\gamma_a$, is $10^{-4}$, and the maximal connection weight, $a_{2,max}$, is 1. The number of video frames, $N$, is taken to be 15. For the parameters in the divisive normalization in Eq 9, $\alpha$ is a small number that avoids zero division and is set to 0.01. The normalization gain, $\beta$, is taken to be three different values, 11, 12, and 13, to investigate the effect of $\beta$.

The modified NBCM rule is also applied on natural images (van Hateren's dataset) with a random spatial jitter and similar results are obtained (see S1 Appendix for detail).

## Measuring spatial phase invariance

Spatial phase invariance, or partial invariance, is one of the most important features of complex cells. Here, sinusoidal gratings with different spatial phases are used as input to the trained model to examine whether model complex cells are invariant to different spatial phases.

**Spatial phase tuning curve.**   First, an exhaustive search for each model complex cell is conducted to find the preferred orientation, spatial frequency, and spatial phase of the sinusoidal grating that evokes the maximal response in the following parameter space: orientation is varied between 0 and 180˚ with steps of 15˚; spatial frequency is varied between 0.05 and 0.4 cycles/pixel with steps of 0.05 cycles/pixel; spatial phase is varied between 0 and 360˚ with steps of 10˚. Then, a sequence of sinusoidal gratings is generated with the preferred orientation and spatial frequency and spatial phases spanning 0–360˚ with a step of 3.6˚ (100 different spatial phases). This sequence of sinusoidal gratings is similar to the drifting sinusoidal gratings used in experimental studies. For each complex cell, the sequence of gratings with different spatial phases is used as the input to the model one after another, while a sequence of responses for each grating is recorded. Therefore, responses vs. spatial phases can be plotted as the spatial phase tuning curve for each complex cell. A complex cell that is completely phase-invariant will have a flat spatial phase tuning curve, while a cell that is phase selective will have a bell-shaped spatial phase tuning curve.

**$F_1/F_0$ ratio.**   Movshon, Thompson, and Tolhurst [3, 7] found that simple and complex cells have different degrees of response modulation when presented with drifting gratings. Subsequently, the degree of response modulation is defined by the ratio $F_1/F_0$ [54], where $F_1$ is the component of the response to the drifting grating at the temporal drifting frequency and $F_0$ is the DC component of the response; i.e., the mean response over time to the drifting grating with spontaneous activity subtracted. Cells are identified as complex if $F_1/F_0 < 1$ and simple if $F_1/F_0 > 1$. Using cell activities in response to drifting gratings, the ratio $F_1/F_0$ is used as a quantitative measure of spatial phase invariance [55]. Since the spatial phase of drifting

gratings changes linearly with time (the speed of change is determined by the temporal frequency of drifting gratings), sinusoidal gratings with different spatial phases are used to mimic the drifting gratings in this study.

## Measuring orientation tuning

Similar to measuring spatial phase tuning, an exhaustive search for each model complex cell is first conducted to find the preferred spatial frequency, orientation, and spatial phase of the sinusoidal gratings that generates maximal complex cell response. Then sinusoidal gratings, with preferred frequency, all possible phases spanning 0–360˚ with step size of 3.6˚, and all possible orientations ranging over 0–180˚ with a step size of 1.8˚, are presented to the model cells. The mean response, $r_k$, to the gratings with all possible phases for each orientation, $\phi_k$, is recorded. The orientations, $\phi_k$, range over 0–180˚ with a step size of 1.8˚. Therefore, there are 100 pairs of $r_k$ and $\phi_k$ that generate the orientation tuning curve for each model complex cell. This orientation tuning curve is then used to compute the preferred orientation and orientation bandwidth as given below.

**The preferred orientation.**   The preferred orientation of a complex cell is the orientation that generates the maximal response. After the orientation tuning curve is obtained, the preferred orientation is simply the peak orientation of the orientation tuning curve.

**Orientation bandwidth.**   The orientation bandwidth used in this paper is the *half-bandwidth*, which was also used in previous experimental studies [56, 57]. Similar to Ringach et al. [57], the orientation tuning curve obtained earlier is first smoothed by a Hanning filter with a half-width of 13.5˚ at half-height. The half-bandwidth is simply the width at $1/\sqrt{2}$ height of the smoothed orientation tuning curve, indicating how broadly the cell is tuned to the preferred orientation. Cells with orientation bandwidth close to 90˚ are invariant to orientations.

## Analyzing complex cells using Nonlinear Input Model

Recently, Almasi et al. [11] applied the Nonlinear Input Model (NIM) framework to analyze RF properties of complex cells and found a diverse range of nonlinear response types in cat V1. Motivated by this study, we employ here the NIM framework to determine the extent to which theoretically learned model complex cells can account for the diverse response properties observed in cat V1.

**Nonlinear Input Model.**   The Nonlinear Input Model (NIM) proposed by McFarland et al. [58] is a general model that assumes minimal but biologically motivated constraints about the underlying neuronal computation; i.e., filters and nonlinearities. The NIM components, when fitted to data, can reveal valuable insights about the mechanism of neural computation.

The structure of the NIM is depicted in Fig 3 and assumes a hierarchical architecture in which the spike rate response, $r$, is a nonlinear function of the input visual stimulus, **s**, given as [58]

$$r = f\left(\sum_{k=1}^{K} g_k(\mathbf{h}_k \cdot \mathbf{s})\right), \tag{10}$$

where $\mathbf{h}_k$ indicates the $k$-th spatial RF filter and $g_k(\cdot)$ denotes the nonlinear function (termed input nonlinearity) applied to the filter's output. The output of each filter is the inner product of the stimulus and each filter ($\mathbf{h}_k \cdot \mathbf{s}$). The model sums inputs from $K$ parallel input streams with arbitrary nonlinearities, which are then passed through a spiking function that gives the firing rate for the cell. The filters and input nonlinearities are described non-parametrically,

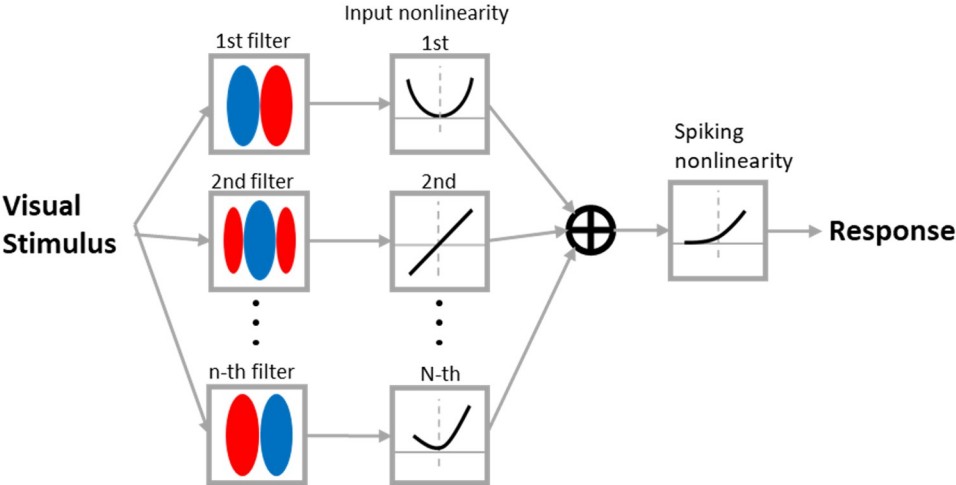

**Fig 3. The structure of Nonlinear Input Model (NIM).** The filter and input nonlinearities determine how it responds to the visual input. The sum of responses of all filters are then passed to a spiking nonlinear function to generate the response of the model.

while the spiking function is assumed to be a monotone-increasing threshold-like function with the form described parametrically as

$$f(x) = \sigma \log\left(1 + \exp\left(\frac{x - \theta}{\sigma}\right)\right) + \delta, \tag{11}$$

where $\sigma$, $\theta$, and $\delta$ are parameters that determine the shape of the function. It should be mentioned that, despite the biological motivation of the NIM, the filters do not necessarily correspond to anatomical cells (subunits) pooled by a complex cell, but rather provide a functional basis that might have possible interpretations. The fitting uses maximum likelihood estimation of the model parameters, including RF filters, input nonlinearities, and spiking nonlinearity, given the stimulus and response. Overall, NIM fitting only assumes a hierarchical structure and a general form of spiking nonlinearity, while all other aspects are estimated from the stimulus (input) and response (output). During the NIM fitting procedure, we assume that the RF structure of each model complex cell comprises two filters and, accordingly, two corresponding input nonlinearities.

**Orientation breadth, spatial frequency breadth, and spatial phase breadth.** Conventional measures of the degree of tuning of V1 cells use a set of standard stimuli, such as sinusoidal gratings, to estimate the tuning bandwidths to characteristics such as orientation, spatial frequency, and spatial phase. In contrast, the characterization used by Almasi et al. [11] estimated the unique set of spatial features to which each cell is sensitive and then estimated the degree of tuning of that cell to those features alone. This was based on an estimate of the nonlinear response to all of these features, with the cell responding strongly to some and weakly to others. The set of spatial features to which a cell is sensitive is identified as follows. Each filter in the NIM corresponds to a spatial feature that is identical in spatial structure to the filter itself. This is a primary feature of the cell and is the spatial feature that the filter is sensitive to when that feature appears embedded in a visual stimulus. As the NIM may have multiple filters, the model cell can be sensitive to multiple spatial features. The set of all features to which the NIM cell is sensitive is given by the feature subspace spanned by the primary features; it encompasses all possible linear combinations of the primary features. For a full interpretation of the NIM, refer to [11].

Based on the nonlinear response of the cell to its subspace of features, the tuning to the orientation, spatial frequency, and spatial phase of these features are defined as follows. First, the orientation, spatial frequency, and spatial phase are calculated for all features in the cell's subspace of features. This is obtained from the peak in the amplitude of the two-dimensional Fourier transform of the feature (see [11]). Next, the feature to which the cell is most sensitive is defined as the feature from the cell's subspace requiring the least contrast to drive the cell at a given reference rate. Finally, tuning breadths are calculated. For example, orientation breadth is defined as the range of orientations of features from the cell's subspace to which the cell showed invariant response. This invariance for a given reference rate includes only features that required less than twice the contrast needed by the cell's most sensitive feature to attain that rate. A similar calculation is applied for the spatial frequency and spatial phase tuning breadths. The maximal values of orientation breadth, spatial frequency breadth, and spatial phase breadth are considered to be 180˚, 0.6 cpd (cycles per degree), and 360˚, respectively. Histograms of these three measures for both model and experimental data, along with the statistical comparison using Welch's t-test, are generated to investigate the performance of the model.

## Results

We present results based on the modified Bienstock, Cooper, and Munro (BCM) and modified Normalized BCM (NBCM) rules. The modified BCM rule learns complex cells that are highly repetitive, while the modified NBCM rule learns different complex cells that are consistent with experimental data.

### Complex cells based on the modified BCM rule

**The model can learn orientation-selective and spatial phase invariant complex cell responses.** One example of model complex cells, C18, is displayed in Fig 4 to illustrate that the model with the modified BCM rule can learn spatial phase invariance as well as orientation selectivity.

In order to show which simple cells provide substantial input to each complex cell, only simple cells with connection weights larger than 0.4 are displayed (the maximal value of the weight is 1). These are referred to as the substantial simple cell inputs. After learning, most connection weights are approaching to either 0 or 1. For these simple cells, the *synaptic field*

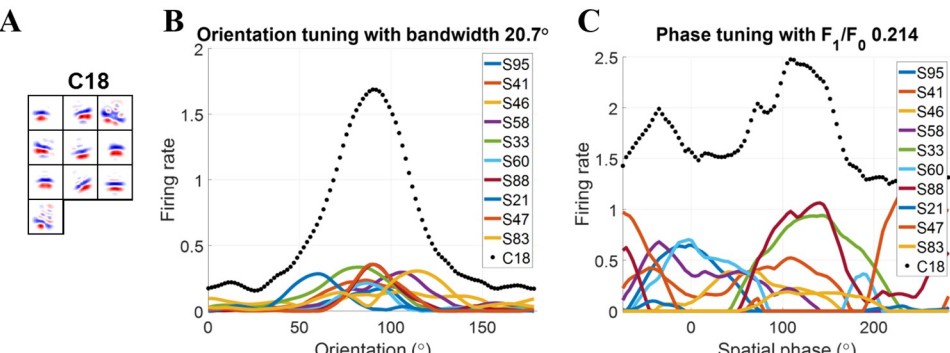

**Fig 4. Complex cell C18 trained using the modified BCM rule with *N* = 15.** (A) Each block is a 16 × 16 synaptic field (defined in Eq 12). Values in each block are normalized to the range [−1 1] when plotting the figure. (B) Orientation tuning curves. (C) Spatial phase tuning curves. Solid lines are for simple cells in the subspace. The dotted line is for complex cell C18. S represents simple cell and the following number is the index of the simple cell.

($\mathbf{S}_f$) [33] is used to visualize the weights between LGN cells and V1 simple cells (Fig 4A), and is defined as

$$\mathbf{S}_f = (\mathbf{A}_{ON}^{u,+} + \mathbf{A}_{ON}^{u,-}) - (\mathbf{A}_{OFF}^{u,+} + \mathbf{A}_{OFF}^{u,-}). \tag{12}$$

These synaptic fields exhibited similarly oriented bands of excitatory ON (red) and OFF (blue) inputs to each other indicating similar orientation tuning of the inputs. This is quantified in Fig 4, which show that, for each complex cell, the substantial simple cell inputs have similar preferred orientations but widely different spatial phase tuning (colored lines). This makes the model cells largely invariant to spatial phase but still selective to orientation (black dotted lines). The spatial phase invariance is quantified by the small values of the $F_1/F_0$ ratio obtained for these examples. A cell that is completely phase-invariant will have $F_1/F_0 = 0$, while a cell that is selective to a small range of spatial phases will have the $F_1/F_0$ ratio close to 2. Smaller values of $F_1/F_0$ indicate greater spatial phase invariance.

**Model complex cells are similar.** While model cells under learning with the modified BCM rule are invariant to spatial phase but selective to orientation, we find that the population of cells are not diverse in their tuning. Instead, they have similar tuning properties because they pool substantial inputs from the same simple cells.

The scatter plot of simple-complex cell connections is shown in Fig 5A, where each dot indicates that the connection weight between the simple and complex cell has a substantial weight ($>0.4$). As seen in Fig 5A, complex cells have substantial connections with the same simple cells, so dots in the scatter plot form vertical lines. However, most simple cells have no substantial connection with any complex cell (90/100), apparent from the complete lack of dots in many columns. Another scatter plot of $F_1/F_0$ vs. preferred orientation is displayed in the right column of Fig 5A, which shows that model complex cells are clustered in a limited region of this space of tuning properties.

The results presented above indicate that the learned model lacks diversity though different model cells are initialized differently, which reflects the lack of competition in the network based on the modified BCM rule.

Similar to the BCM rule, the modified BCM rule does not introduce any competition between cells, so the learned cells exhibit similar tuning properties, as discussed by Willmore et al. [45]. Therefore, we introduce soft competition using the modified NBCM to learn different tuning properties for model complex cells.

## Complex cells based on the modified NBCM learning rule

**A diverse range of tuning properties of complex cells across the population are learned.** The NBCM rule adds divisive normalization of responses to the BCM rule [46]. This re-scales the neuronal responses based on responses of other units in the model, so a form of soft competition between cells is introduced to the model and leads to the model learning different cells.

In comparison with model complex cells learned by the modified BCM rule, the modified NBCM rule forces the model to learn different complex cells with different orientations such that there is little overlap or repetition in substantial simple cell inputs across the complex cell population. The scatter plots of simple-complex cell connections with different values of normalization gain, $\beta$, are given in the left column of Fig 5B–5D, which shows that simple-complex connections are considerably more diverse than those given by the modified BCM learning rule (Fig 5A). Dots in the figure appear more randomly assigned to complex cells, and there is much less evidence of vertical lines, such as those that appear in Fig 5A. Furthermore, the scatter plots of $F_1/F_0$ vs. preferred orientation (right column of Fig 5B–5D) cover a much

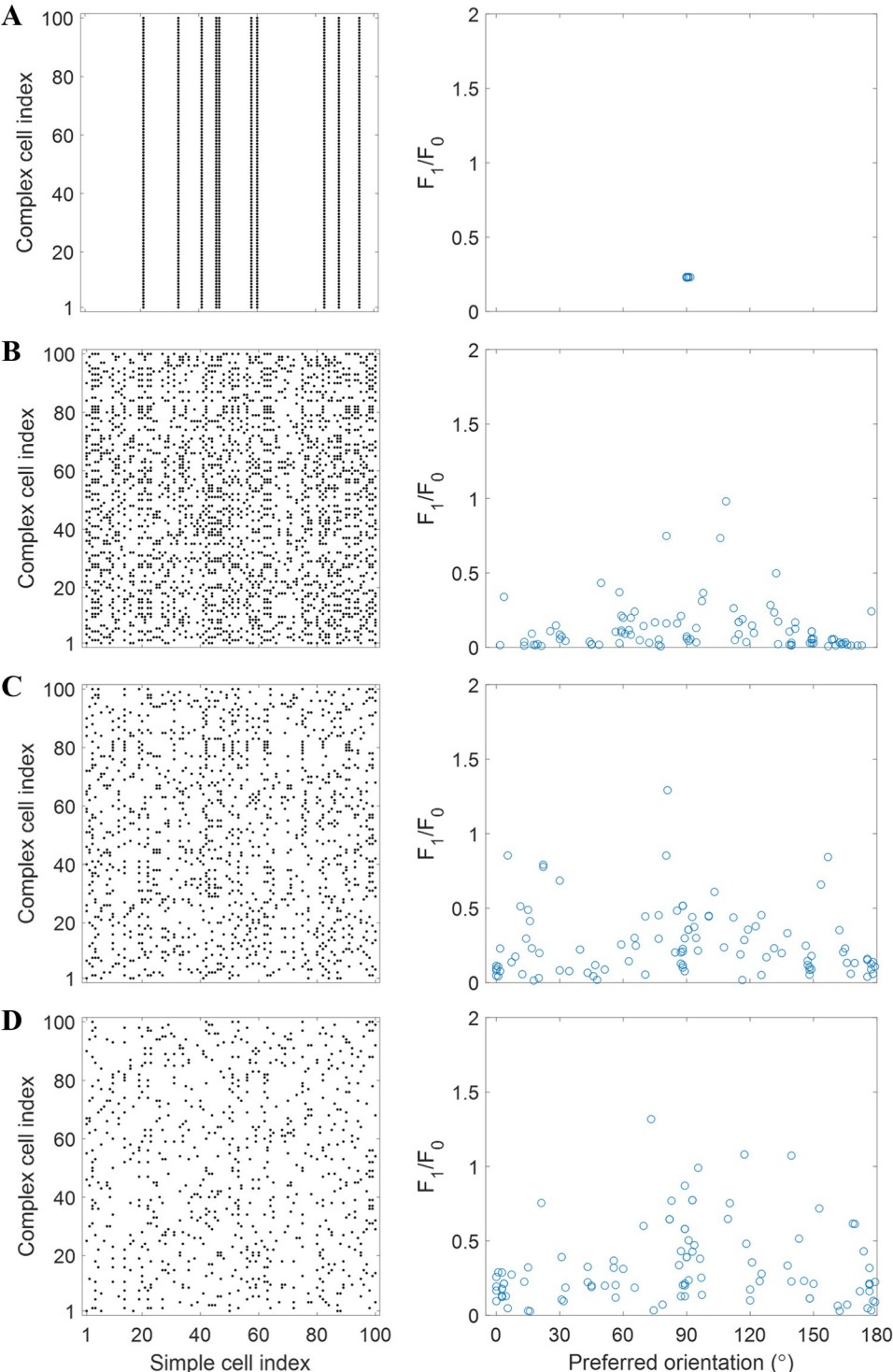

**Fig 5. Scatter plots that investigate the diversity of learned complex cells.** Left: scatter plot of simple-complex cell connections for the model. The dots in each row represent the indices of simple cells that have substantial weights (>0.4) with the complex cell indicated by an index on y-axis. Right: scatter plot of $F_1/F_0$ vs. preferred orientation for all model complex cells. (A) Modified BCM rule. (B)-(D) Modified NBCM rule with $\beta = 11$ (B), $\beta = 12$ (C), and $\beta = 13$ (D).

wider range compared with Fig 5A. They include all orientations (0 to 180˚) and a range of spatial phase tuning from complete invariance ($F_1/F_0$ close to 0) to moderate tuning ($F_1/F_0$ between 0 and 1). This indicates that the tuning properties are more diverse than those obtained with the modified BCM rule.

**Normalization introduces smooth competition required to learn different complex cells.** The competition between neurons is important for the network to learn diverse properties. Results presented above indicate that the competition introduced by the normalization in the NBCM rule promotes the diversity of tuning across the population of complex cells. However, based on the hierarchical assumption of the model, the competition introduced by efficient coding is too strong to produce RF properties of complex cells (see Discussion and S2 Appendix). This is also consistent with the result that the normalization gain, $\beta$, determines the level of competition.

Fig 5 shows that the modified NBCM rule can learn different complex cells that have small $F_1/F_0$ ratio and different preferred orientation tuning. Moreover, simple-complex connections become sparser as the normalization gain, $\beta$, increases.

The right column of Fig 5B–5D indicates that more complex cells have smaller $F_1/F_0$ ratio when $\beta$ is small. This observation is supported by the histogram of $F_1/F_0$ shown in Fig 6, which shows that the distribution of $F_1/F_0$ skews towards zero when $\beta$ decreases, indicating greater spatial phase invariance. The histogram of model complex cells with $\beta = 12$ (Fig 6C) has a close resemblance to experimental data (Fig 6A).

Furthermore, we also investigate the diversity of how tightly or broadly cells are tuned to orientation in the population of model complex cells. This is assessed by measuring orientation

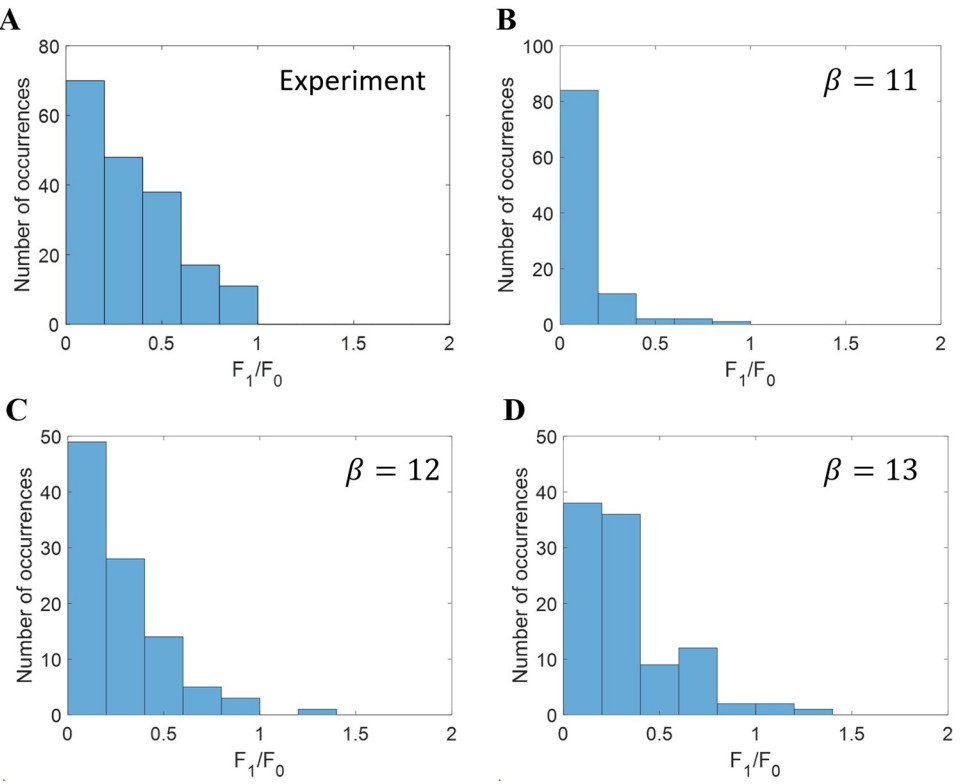

**Fig 6. Histograms of $F_1/F_0$ for models based on the modified NBCM rule.** (A) Experimental complex cells [57]. Model complex cells learned with (B) $\beta = 11$, (C) $\beta = 12$, and (D) $\beta = 13$.

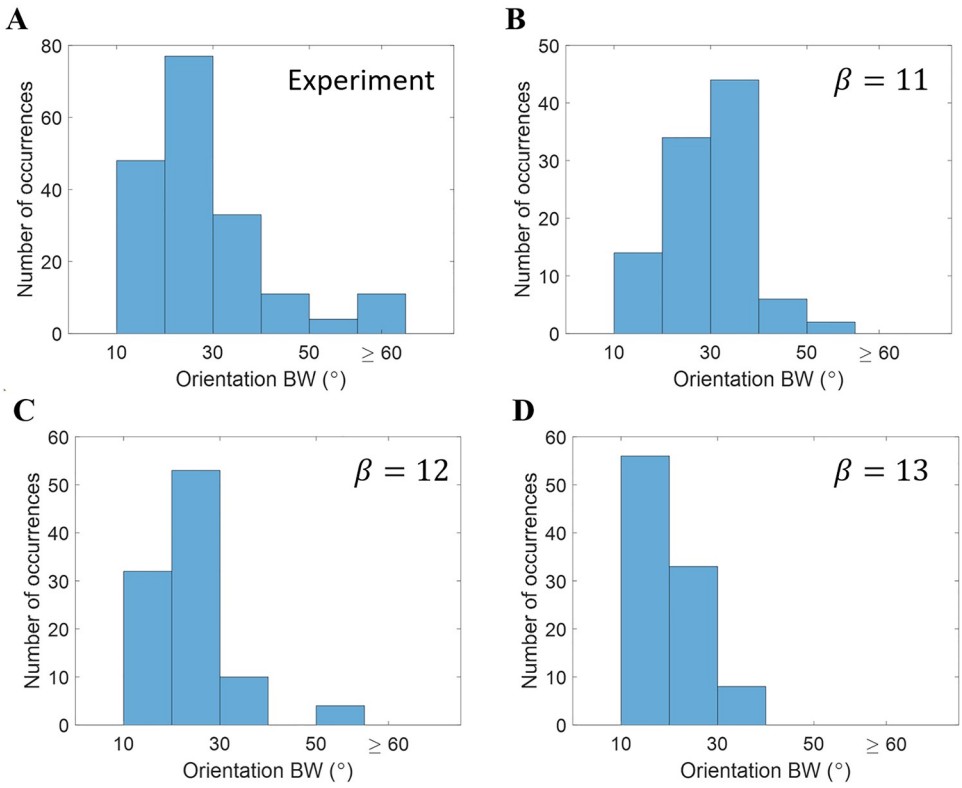

**Fig 7. Histograms of half-bandwidth for models based on the modified NBCM rule.** (A) Experimental complex cells [57]. Model complex cells learned with (B) $\beta = 11$, (C) $\beta = 12$, and (D) $\beta = 13$.

tuning bandwidth of responses to drifting gratings (see for detail). Fig 7 shows the histogram of half-bandwidth for experimental and model data (only data points with $F_1/F_0 < = 1$ are included). The histograms in Fig 7 show that the distribution of orientation bandwidth skews towards smaller values as $\beta$ increases. Model complex cells have a similar proportion of cells with wide orientation bandwidth ($\geq 60°$), but models with different values of $N$ change the proportion of cells with small orientation bandwidth. The histogram of model complex cells with $\beta = 12$ (Fig 7C) matches with the experimental data better than other values of $\beta$. However, experimental data has a relatively smooth transition from low orientation bandwidth to high orientation bandwidth, while model data has less variability in the region from 40° to 60°. The discrepancies between experimental and model data are reviewed in Complex cells based on the modified NBCM learning rule.

Combining Figs 5–7, we conclude that the normalization gain, $\beta$, determines the level of competition between model complex cells. As $\beta$ decreases, there is less competition and each complex cell pools more simple cells, which leads to a larger spatial phase invariance (smaller $F_1/F_0$ ratio) and wider orientation tuning (larger orientation bandwidth). Therefore, smooth competition that promotes diversity but preserves some invariance is crucial to the learned RF properties of complex cells.

Given the better match to experimental data (Figs 6C and 7C), the data set for $\beta = 12$ with the modified NBCM learning rule is used for further analysis in the following section.

**Examples of model complex cells.** Some examples of complex cells are provided to demonstrate the diversity of model complex cells and the resemblance to experimental complex cells.

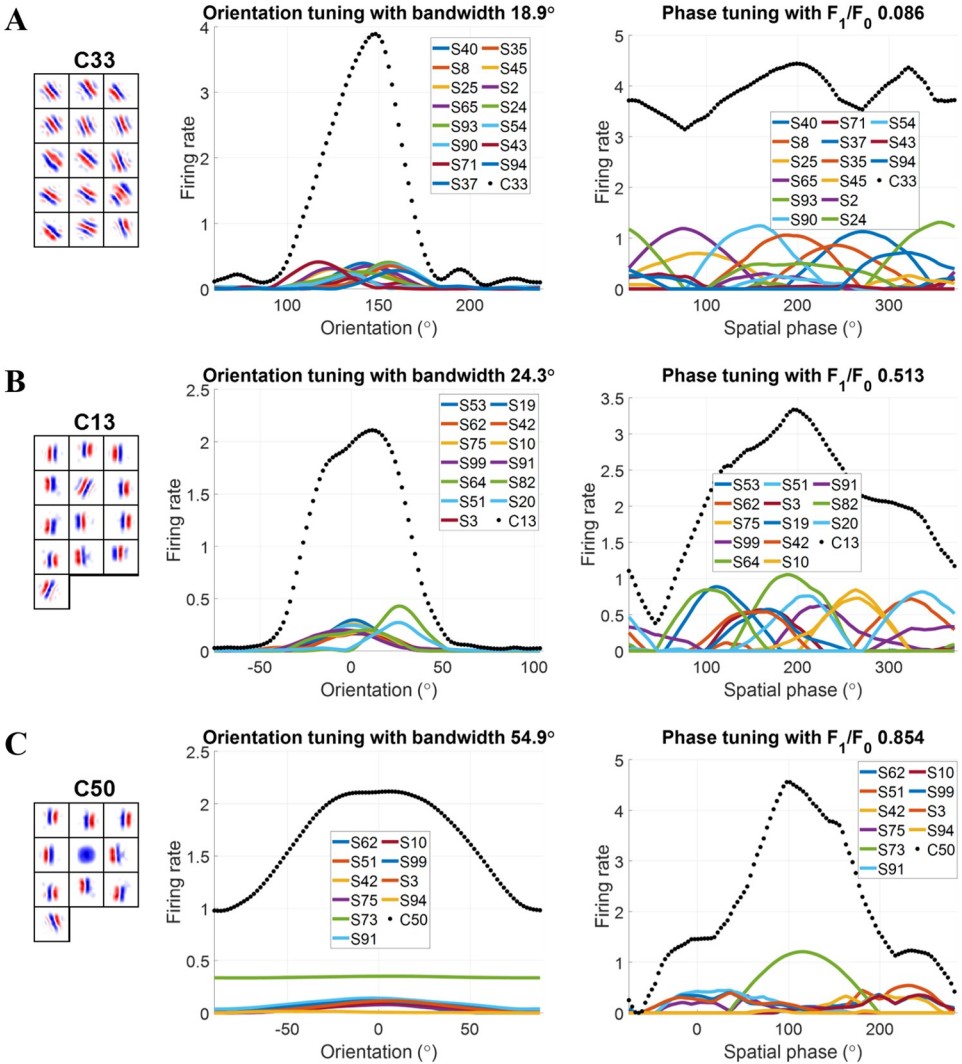

**Fig 8. Examples of model complex cells based on the modified NBCM rule.** Left: each block is a 16 × 16 synaptic field (defined in Eq 12) for simple cells in the subspace and values in each block are normalized to the range [−1 1] when plotting the figure. Middle: orientation tuning curve. Right: spatial phase tuning curves. Solid lines are for simple cells in the subspace. Dotted line is for complex cell. S represents simple cell and the following number is the index of the simple cell. (A) Complex cell that is invariant to all spatial phases. (B) Complex cell that shows invariance to perturbations in orientation. (C) Complex cell that is invariant to orientation but not spatial phase.

*A complex cell that is invariant to all spatial phases*: Fig 8A shows a model complex cell (C33) that shows a high degree of invariance across all spatial phases, as its response shows limited modulation with spatial phase. Complex cell C33 has substantial simple cell inputs with similar but not identical orientations and spatial phase tuning curves covering all phases. This leads to spatial phase invariance for the model complex cell with a very small $F_1/F_0 = 0.086$. The model complex cell is highly selective to a particular orientation, as seen in the orientation tuning curve. Qualitatively similar cells are observed in cat primary visual cortex [11].

*A complex cell that shows invariance to perturbations in orientation*: Fig 8B shows a model complex cell (C13) that has two major orientations in the subspace: one vertical and another oblique. As a result, the orientation tuning curve has a wider bandwidth compared with

complex cell C33 in Fig 8A. Complex cell C13 is invariant to a limited range of spatial phase because the spatial phase tuning curves of simple cells in the subspace only cover a subset of 360˚. Qualitatively similar cells are observed in cat primary visual cortex [11].

*A complex cell that is invariant to orientation but not spatial phase*: Fig 8C shows an model complex cell (C50) that is invariant to orientation but not to spatial phase. Simple cells in the subspace have various orientations including vertical, oblique, and non-oriented simple cells. Qualitatively similar cells are observed in cat primary visual cortex [11].

**Population statistics compared with experimental data.** The example model cells described above suggest that there is a diversity of tuning for orientation and spatial phase in the model population. This is consistent with a recent study that characterized nonlinear RF models in a population of cat V1 neurons, and also found a diversity of tuning properties [11]. To quantitatively compare our results for the learned model with these experimental results, population statistics are analyzed using the three measures of tuning used in the experimental study (summarized in Materials and methods): orientation breadth, spatial frequency breadth, and spatial phase breadth. Comparisons between model and experimental data [11] are shown in Fig 9.

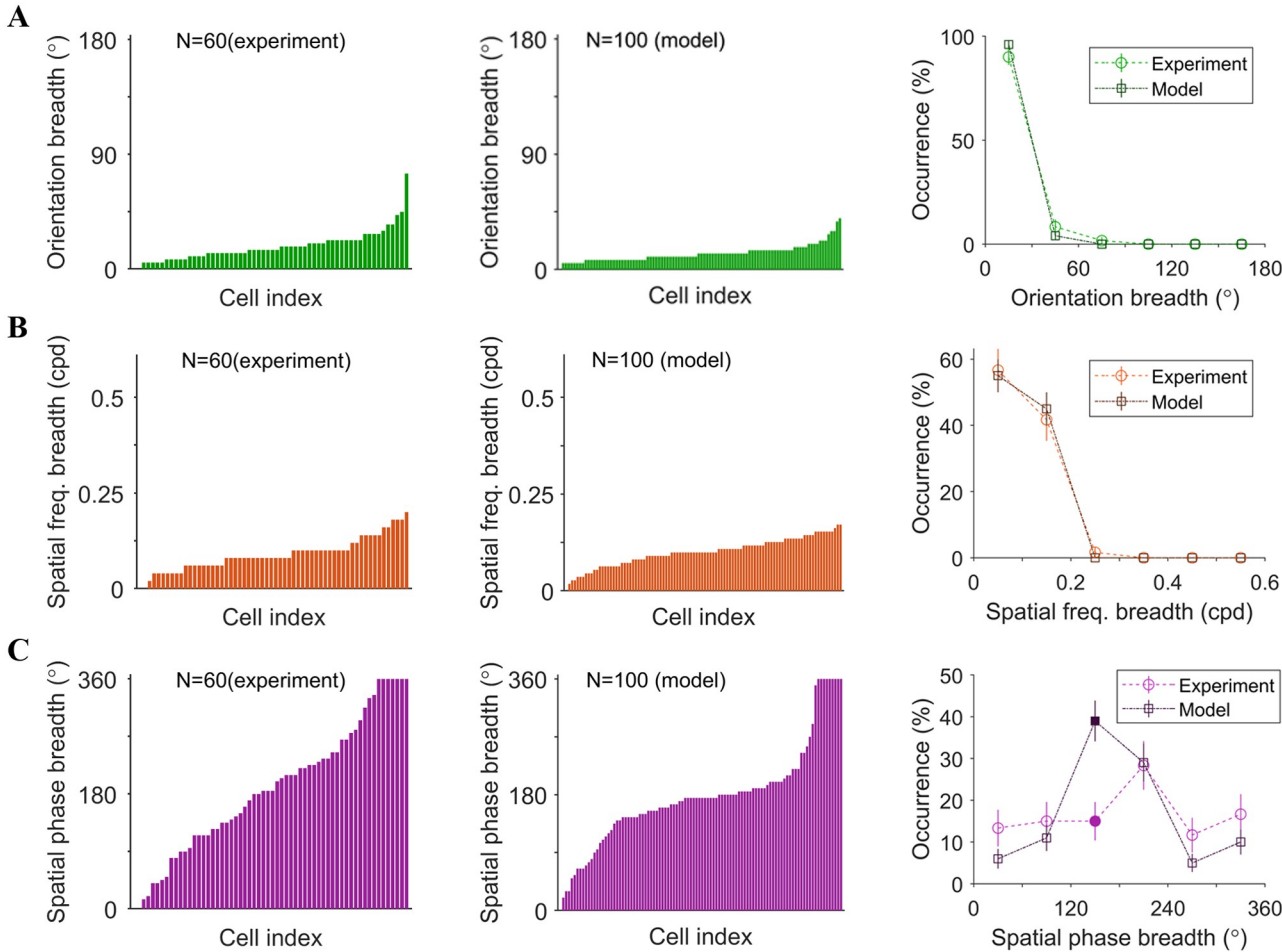

**Fig 9. Comparison between experimental data [11] (left) and model data trained using the modified NBCM rule (middle).** Right: histograms, where filled points indicate differences that are significant (p-value<0.05; Welch's t-test). (A) Orientation breadth (˚). (B) Spatial frequency breadth (cycles per degree). (C) Spatial phase breadth (˚).

Fig 9A shows that most cells for both model and experimental data are highly tuned to orientation (orientation breadth <45˚), but experimental complex cells have a somewhat broader distribution of orientation breaths than the model complex cells. In general, the overlap between the histograms of model and experimental data accounts for 88.7% of the histograms spanned by both model and experimental data.

Fig 9B shows that model data has similar range of tuning to spatial frequency as experimental data. The spatial frequency breadth of the model displayed in the figure is scaled to match the experimental data. It is important to mention that the spatial frequency of the model depends on the assumption of how large the visual field the input image represents. Therefore, different sizes of the visual field will scale the spatial frequency. In general, the overlap between the histograms of model and experimental data accounts for 93.6% of the histograms spanned by both model and experimental data.

Fig 9C shows that both model and experimental data cover a wide range of spatial phase tuning, except that the model data has more complex cells with spatial phase breadth around 150˚. In general, the overlap between the histograms of model and experimental data accounts for 60.4% of the histograms spanned by both model and experimental data.

Overall, the model can account for the diversity of complex cells found in the experimental study of Almasi et al. [11]. Despite some discrepancies in the histograms of these three measures of tuning, the model can capture the trends of the distributions.

## Discussion

### Natural video vs. jittered natural images

Complex cells are trained on the natural video in this paper and jittered natural images in S1 Appendix. The spatio-temporal information contained in consecutive frames of either a natural video or jittered patches of a natural image allows the model to learn complex cells that match experimental data well. Consecutive frames of a video contain visual information in this moving world, while jittered images generate visual inputs that are similar to eye saccades.

### The modified BCM rule vs. the modified NBCM rule

Both the modified BCM and modified NBCM plasticity rules can learn the input weights of model complex cells that result in spatial phase invariance and orientation selectivity. However, the complex cells that learned using the modified BCM rule were highly similar to each other in their tuning properties, while complex cells that learned with the modified NBCM rule exhibited a wide diversity of tuning to orientation and spatial phase. The similarities between the distributions of tuning properties in experimental and model data learned by the modified NBCM rule suggest that this rule can explain complex cell properties well and that complex cells can be learned in a biologically plausible neural network model.

### Response normalization

Response normalization was implemented in the modified NBCM model as a form of divisive gain modulation dependent on the activity of all other neurons in the network. It plays an important role in the model, enabling it to learn complex cells with different tuning because it introduces competition between cells that discourages cells from responding concurrently. Without normalization, model complex cells tend to have similar tuning and lack the diversity seen in experimental data. Normalization of responses in Eq 9 was first proposed by Heeger [46] and is suggested to be a canonical neural computation, for which there is strong evidence [47]. While the explicitly divisive form of response normalization proposed by Heeger is

difficult to justify biophysically at the level of a single neuron, Rubin et al. [49] suggested that normalization can be implemented in the cortex using a stabilized supralinear network that employs recurrent connections. The results presented in this study indicate that the model can learn complex cells using a hierarchical structure, but this does not rule out the importance of the recurrent structure because recurrent connections might be important to implement normalization in the neural circuits.

For the normalization in our model (Eq 9), $\alpha$ is a small positive constant in the denominator that avoids zero division in normalization. If $\alpha$ is large compared with $\Sigma(r^C)^2$, the normalization is simply a constant scaling of the neural response. Therefore, we choose a small value, $\alpha = 0.01$. Other small values of $\alpha$, such as $10^{-3}$, will generate similar results (see an example in S1 Fig). The value of $\beta$ controls the magnitude of responses following the normalization. When $\beta$ is large, learned complex cells pool fewer simple cell inputs and thus are less invariant to spatial phase. In other words, $\beta$ controls the level of competition between cells and large values lead to stronger competition such that learned cells are more distinctly tuned. The interaction between normalization and the learning rule of the model could potentially account for more experimental data with different values of the parameters.

## Discrepancies between model and experimental data

The model proposed in this paper can pool simple cell inputs into the subspace of complex cells. However, although the main features of the model agree qualitatively with the experimental data, some discrepancies between the model and experimental data exist and remain to be explored.

The model can account for the diversity of tuning properties of complex cells in the cat visual cortex reported by Almasi et al. [11], but there are some differences in the distributions of the population statistics. In addition, the experimental data of Ringach et al. [57] shows greater diversity in orientation bandwidths than the model data. This may be because the model cells are only a subset of cortical cells. Alternatively, we cannot exclude the possibility that choices of free parameters in our model might lead to results with a greater range of orientation bandwidths. Another reason for differences in orientation bandwidths between the model and experimental data may be related to the visual stimuli and the temporal dynamics of the neural system. In the experimental study that measured orientation selectivity in macaque V1 [57], drifting sinusoidal gratings were used as the visual stimuli, which engage the temporal dynamics of the cells. However, when calculating the spatial phase tuning properties of model cells, steady-state responses to each spatial phase of the drifting gratings were used. In addition, the current model does not incorporate the temporal dynamics of cells. The investigation of a model that incorporates temporal dynamics is left for future research.

## A recent model based on predictive coding

Franciosini et al. [59] recently presented modeling work of complex cells based on predictive coding. The difference from the model proposed here is that their model used symmetric connections between simple cell inputs and simple cells, and between complex cell inputs and complex cells. Furthermore, there are two stages of pooling between simple cell responses and the complex cell inputs: spatial max-pooling with kernel size $2 \times 2$ and group-pooling between neighbouring groups. After learning the connections between simple cell inputs and simple cells, and between complex cell inputs and complex cells, their model can learn a topographical map of simple cells such that model complex cells pool local simple cell receptive fields that are similar in tuning to orientation but variable in tuning to spatial phase to achieve spatial phase invariance. This is a plausible mechanism in animals, such as felines and primates, that have

such topographic orientation maps, but does not apply to animals that lack these maps in visual cortex, such as rodents and lagomorphs. The learning of complex cells depend on the pooling mechanism, but how the pooling stages in their model are implemented in a biologically plausible way is not clear. Another limitation of the model is that the learning rule is not local and the presumed symmetric connections are not biologically realistic.

### Efficient coding

The principle of efficient coding and its associated learning rule helps the model units learn features, which also makes them highly selective to their preferred features. The competition brought about by efficient coding is indispensable for achieving selectivity and diversity of responses, either through feedback [33, 60] or lateral connections [50, 61]. However, the competition might also be very strong, as can be seen from Fig 11 of Olshausen and Field [53], where the feedforward response is much stronger than the response in the efficient coding model.

Based on our assumption of a hierarchical structure, we conducted extensive investigation to understand if efficient coding, implemented as a sparse coding model, can learn the correct simple cell inputs to complex cells by finding an efficient representation of natural stimuli with temporal information. We found that model complex cells can learn to pool simple cell inputs with a wide range of spatial phase selectivities. However, the strong competition between complex cells introduced by efficient coding to make responses sparse suppresses responses of model complex cells to all but a small set of spatial phases. As such, they behave like simple cells that are very selective to spatial phase (see S2 Appendix for details).

However, the BCM-based models proposed in this paper can learn receptive field properties of complex cells. The modified BCM rule can learn useful representations and the response normalization introduces competition to the network. In terms of competition, response normalization is much softer than efficient coding because efficient coding pushes the activity of many cells to zero while normalization only adjusts the levels of activities. Therefore, the soft competition introduced by response normalization helps the model learn different complex cells while still allowing broad tuning to simple cell inputs with different spatial phase selectivities.

### Limitations and future work

The learning model of complex cells proposed here can explain the emergence of receptive field properties of experimentally recorded complex cells. However, some limitations remain in the current model and can be improved further. First, the training processes of simple and complex cells are separate; i.e., simple cells are trained first and then complex cells. Second, the complex model has no temporal dynamics (although the simple cell model does). Finally, response normalization in this paper does not have an explicit biologically plausible implementation. For future work, a more unified model with temporal dynamics and normalization can be incorporated into the model to account for complex cell responses, which may potentially reduce the discrepancies between model and experimental data as well.

## Supporting information

**S1 Appendix. Complex cells can also be learned using jittered images.** This document gives details on learning complex cells using jittered natural images instead of a natural video. (PDF)

**S2 Appendix. Can complex cells be learned by efficient coding?.** This document gives details on investigating efficient coding for learning complex cells.
(PDF)

**S1 Fig. Scatter plots that display the diversity of learned complex cells based on the modified NBCM with $\alpha = 10^{-3}$ while other parameters are the same as Fig 5C.**
(TIF)

## Acknowledgments

The authors thank Michael Ibbotson for his helpful insights.

## Author Contributions

**Conceptualization:** Yanbo Lian, Anthony N. Burkitt, Hamish Meffin.

**Data curation:** Yanbo Lian, Ali Almasi.

**Formal analysis:** Yanbo Lian, Ali Almasi, Hamish Meffin.

**Funding acquisition:** Anthony N. Burkitt, Hamish Meffin.

**Investigation:** Yanbo Lian, Anthony N. Burkitt, Hamish Meffin.

**Methodology:** Yanbo Lian, Ali Almasi, David B. Grayden, Tatiana Kameneva, Anthony N. Burkitt, Hamish Meffin.

**Project administration:** Yanbo Lian, Anthony N. Burkitt, Hamish Meffin.

**Resources:** Ali Almasi, David B. Grayden, Tatiana Kameneva, Anthony N. Burkitt, Hamish Meffin.

**Software:** Yanbo Lian, Ali Almasi, Hamish Meffin.

**Supervision:** Anthony N. Burkitt, Hamish Meffin.

**Validation:** Yanbo Lian, Anthony N. Burkitt, Hamish Meffin.

**Visualization:** Yanbo Lian, Ali Almasi, Hamish Meffin.

**Writing – original draft:** Yanbo Lian.

**Writing – review & editing:** Yanbo Lian, Ali Almasi, David B. Grayden, Tatiana Kameneva, Anthony N. Burkitt, Hamish Meffin.

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
