## [Decision Letter · Decision Letter 0]

27 Jul 2020

Dear Dr. Lian,

Thank you very much for submitting your manuscript "Learning receptive field properties of complex cells in V1" for consideration at PLOS Computational Biology.

As with all papers reviewed by the journal, your manuscript was reviewed by members of the editorial board and by several independent reviewers. In light of the reviews (below this email), we would like to invite the resubmission of a significantly-revised version that takes into account the reviewers' comments.

The reviewers raised several concerns. As both point out, it is crucial that model data is compared to experimental data quantiatively with the appropriate statistical tests. Otherwise the gained biological insight remains (too) vague. You also should clarify the presentation of your model and do the additional test with the movie data set suggested by reviewer 2. I understand that this will be a substantial amount of work, and eventual acceptance is far from guaranteed (it is crucial that the biological insight gained is judged as substanital), but I wanted to give you the opportunity to perform such a major revision.

We cannot make any decision about publication until we have seen the revised manuscript and your response to the reviewers' comments. Your revised manuscript is also likely to be sent to reviewers for further evaluation.

Sincerely,

Wolfgang Einhäuser

Deputy Editor

PLOS Computational Biology

Reviewer's Responses to Questions

**Comments to the Authors:**

Reviewer #1: In this paper, authors propose a model for learning receptive field properties of complex cells in V1. Compared to existing models, they use a biologically motivated model which leads to interesting results, and which are compared to experimental observations. The paper is quite clearly written and organised, though I found some minor typographic errors. I recommand a major revision in order to clarify the message message of the paper and improve its impact to the coimmunity.

The first major point will be to clarify clarify the quantitative comparison of the results from the model with the biological data. In particular, you should identify some key aspects of simple versus complex cells and show what parameters are essential for obtaining a good fit. Moreover, from the analogy with the NIM and the results by Almassi, How would you justify some aspects of your model with the results obtained in your model (notably the non-linearities) ?

Another major point will be to simplify the presentation of the model by summarizing the different heuristics, and to highlight the most important factors which lead to the emergence of complex cells properties in your neuronal model. For instance, it seems that the homeostasis mechanism is that you introduce in your modified BCM whole learning rule is very important to obtain a realistic result as shown in the comparison between figures 8 and 9. The parameters of this modified rule must be quantitatively explored to check to check at what point you switch from one state (Fig 8) to the other (Fig9). As such￼, there is an important link make with efficient coding (“The learning rule derives from efficient coding") which is well discussed  and which may help guide the presentation of the model.

Also, make sure the code will be made available at publication time.

minor points:

L104 :« « Hos »a »« > « Ho »oya »

L125 : « we simply use a sequences of natural images «   > « we simply use a sequence of natural images "

L196 "it p more emphasis " > "it puts more emphasis " ?

L208 "plasticit "

L298 "τL and τS, for LGN and simple cells are taken to be 10 ms " - both are equal?

L318 "The learning rate, η1, is 3 " - this value seems very high compared to other studies...

L375 : check syntax of the long line ending with ", are presented to the model cell "

Fig 10 A instead of "expt" write "experiment"

Reviewer #2: This manuscript analyzes how responses properties of complex cells can be learned from the inputs of simple cells in the primary visual cortex. There is an important aspect in complex cell responses that has not received much attention so far, namely that their responses show less invariance to spatial phase as is assumed in current idealized theories of visual cortex. The authors that this diversity can be accounted by only when connections are learning using a modified BCM rule that includes normalization of neural responses. The manuscript can make an important contribution to the field but needs to be revised substantially in the following ways:

1) Comparison with experimental histograms is currently very qualitative. This comparison needs to be done with statistical tests

2) The orientation tuning width for complex cells is measured as a variance in the preferred orientation of subunits. However, there is also finite orientation tuning for each subunit. This orientation tuning width can be extracted by taking a Fourier transform of the spatial profile of each subunit. Example of how this has been done can be found in Sharpee, Miller, & Stryker J Neurophysiology 2008

3) It would be better to use natural videos rather than jittered natural images. Such videos are part of the van Hateren dataset.

4) There are too many figures in the manuscript. For example, Figure 1, 2, 3 and 5 might be removed without loss.

There are typos on line 208 and 196.

**Have all data underlying the figures and results presented in the manuscript been provided?**

Reviewer #1: Yes

Reviewer #2: Yes

PLOS authors have the option to publish the peer review history of their article (what does this mean?). If published, this will include your full peer review and any attached files.

Reviewer #1: **Yes: **Laurent U Perrinet

Reviewer #2: No
---

## [Decision Letter · Decision Letter 1]

1 Dec 2020

Dear Dr. Lian,

Thank you very much for submitting your manuscript "Learning receptive field properties of complex cells in V1" for consideration at PLOS Computational Biology. As with all papers reviewed by the journal, your manuscript was reviewed by members of the editorial board and by several independent reviewers. The reviewers appreciated the attention to an important topic. Based on the reviews, we are likely to accept this manuscript for publication, providing that you modify the manuscript according to the review recommendations.

Sincerely,

Wolfgang Einhäuser

Deputy Editor

PLOS Computational Biology

[LINK]

Reviewer's Responses to Questions

**Comments to the Authors:**

Reviewer #1: Many thanks to the authors for their revised manuscript and for their responses to the comments from the reviewers. The manuscript is now much improved. There are still some glitches in the presentation of the results which may hinder conveying the global message. I would recommend a minor revision prior to acceptance.

First, the paper is in general clearly written and would benefit for some simplification in the presentation of the results. First, I recommend to fully check the syntax. In particular, some sentences are more than three lines long and should be split. There a number of occurrences of missing articles (eg l550 " of response " ; l573 "Complex cell that" > "A complex cell that" ; l646 " introduces competition" ; ...). Also some terms could be better chosen, for instance l400 "how widely" -> "how broadly" (?); l487 "the *pronounced* simple cell"; l534 "moderate tuning ".

Second, the main contribution of the paper is to propose a novel model and to show how it models the learning of complex cells. instead of being "yet another model" it would be a great contribution to show the generality of their result and highlight the novelty in that model. In particular, having as a result that $\\beta \\approx 12$ does not bring much to the community. Showing that having a smooth competition (through normalization) allows to have a better fit should be discussed. Another point is that saying that "efficient coding does not learn complex cells" is quite overstated. Efficient coding has many facets and in appendix S1, you study just that relative to your architecture.

minor:

l487 "Figs > Fig

**Have all data underlying the figures and results presented in the manuscript been provided?**

Reviewer #1: Yes

PLOS authors have the option to publish the peer review history of their article (what does this mean?). If published, this will include your full peer review and any attached files.

Reviewer #1: **Yes: **Laurent U Perrinet
---

## [Decision Letter · Decision Letter 2]

9 Feb 2021

Dear Dr. Lian,

We are pleased to inform you that your manuscript 'Learning receptive field properties of complex cells in V1' has been provisionally accepted for publication in PLOS Computational Biology.

Best regards,

Wolfgang Einhäuser

Deputy Editor

PLOS Computational Biology

Reviewer's Responses to Questions

**Comments to the Authors:**

Reviewer #1: Authors have correctly responded to my minor comments.

I encourage you to further study the generality of your proposed model and the function of the apparent division between simple and complex cells within a unified theory. Congratulations for this final manuscript !

**Have all data underlying the figures and results presented in the manuscript been provided?**

Reviewer #1: Yes

PLOS authors have the option to publish the peer review history of their article (what does this mean?). If published, this will include your full peer review and any attached files.

Reviewer #1: **Yes: **Laurent Perrinet

---

## [Editor Report · Acceptance letter]

23 Feb 2021

PCOMPBIOL-D-20-00804R2 

Learning receptive field properties of complex cells in V1

Dear Dr Lian,

I am pleased to inform you that your manuscript has been formally accepted for publication in PLOS Computational Biology. Your manuscript is now with our production department and you will be notified of the publication date in due course.

With kind regards,

Alice Ellingham
